# Effect of Associated Bacteria GD1 on the Low-Temperature Adaptability of *Bursaphelenchus xylophilus* Based on RNA-Seq and RNAi

**DOI:** 10.3390/microorganisms11020430

**Published:** 2023-02-08

**Authors:** Yuchao Yuan, Min Pan, Luyang Shen, Yuqian Liu, Qinping Zhu, Jingxin Hong, Jianren Ye, Jiajin Tan

**Affiliations:** Co-Innovation Center for Sustainable Forestry in Southern China, College of Forestry, Nanjing Forestry University, Nanjing 210037, China

**Keywords:** *Bursaphelenchus xylophilus*, associated bacteria, low-temperature adaptability, RNA-seq, RNAi

## Abstract

To explore the effect of associated bacteria on the low-temperature adaptability of pinewood nematodes (PWNs), transcriptome sequencing (RNA-seq) of PWN AH23 treated with the associated bacterial strain *Bacillus cereus* GD1 was carried out with reference to the whole PWN genome. Bioinformatic software was utilized to analyze the differentially expressed genes (DEGs). This study was based on the analysis of DEGs to verify the function of *daf-11* by RNAi. The results showed that there were 439 DEGs between AH23 treated with GD1 and those treated with ddH_2_O at 10 °C. There were 207 pathways annotated in the KEGG database and 48 terms annotated in the GO database. It was found that after RNAi of *daf-11*, the survival rate of PWNs decreased significantly at 10 °C, and fecundity decreased significantly at 15 °C. It can be concluded that the associated bacteria GD1 can enhance the expression of genes related to PWN low-temperature adaptation and improve their adaptability to low temperatures.

## 1. Introduction

Pinewood nematodes (*Bursaphelenchus xylophilus*) are a global forest pest that have caused devastating damage to pine trees in many countries in Asia, Europe and other continents [1,2]. As PWNs spread, they continuously adapt to the environment, and there is a trend of PWN invasion breaking through the northern boundary of the suitable growth zone, spreading to higher latitudes with lower temperatures [3,4,5].

Cold tolerance refers to the survival ability of organisms exposed to low temperatures and exists in most organisms [6], including nematodes [7]. The survival and reproductive ability of PWNs are affected at low temperatures [8], where PWNs enter a state of diapause, thus improving their ability to survive these temperatures [9]. The survival and reproductive ability of PWNs are often used as indicators to quantify low-temperature adaptability [8]. To adapt to the cold environment, physiological and biochemical reactions occur in the nematodes, and many lipid droplets and sugars accumulate. These reactions also conserve resources for the subsequent growing season, and fatty acids are components of lipids. Additionally, this accumulation can reduce freezing damage to nematodes [10]. Furthermore, glycerol and trehalose may contribute to the low-temperature adaptability of nematodes [11].

Studies investigating the relationship between the presence of *Meloidogyne* spp. and severity of blight have revealed that mixed inoculation with *Pseudomonas solanarearum* and *Meloidogyne* spp. leads to faster and more severe blight [12,13]. Sergio et al. found that tobacco root-knot nematode disease is closely related to interactions between soil microbial communities [14]. Subsequently, scholars have reported five key microorganisms that affect the occurrence of tobacco root-knot nematode disease—*Pseudomonas*, *Bryobacter*, *Variibacter*, *Coniochaeta* and *Metarhizium* [15,16]—illustrating the role of bacteria that accompany nematode diseases. The role of bacteria in pine wilt disease (PWD) has been extensively investigated. An initial study of the toxin of PWNs revealed that it might be related to *Pseudomonas* spp. [17]. A year later, Tada et al. isolated toxic bacteria from pinewood nematodes [18]. *Bacillus cereus*, *Bacillus subtilis* and *B. megaterium* were isolated from PWNs and were harmful to the black pine callus, suspension cells and seedlings [19]. Phenylacetic acid produced by the associated bacteria also showed strong phosphine activity [20,21]. Li et al. performed inoculation experiments after disinfecting the surface of nematodes and found that the incidence of PWD caused by nematodes not sterilized was faster and more severe [22]. This finding is consistent with Vicente’s results [23]. However, some scholars have drawn different conclusions using vaccination experiments. Bolla found that pine wilting could still occur after treating pinewood nematodes with antibiotics [24]. Tamura obtained similar results in experiments using sterile PWN inoculation with pine [25]. To date, many studies have shown that the bacteria associated with PWNs can significantly improve their ecological adaptability during spreading, such as their antioxidant [26], anti-terpene toxicity [27] and alcohol adaptation [28] abilities. Concerning the current research progress, associated bacteria play a more crucial role in the occurrence of PWD. During PWN infestation, they may reduce the host’s resistance or enhance the adaptability of the nematodes.

The transcriptome is the sum of all transcripts under a specific environment. Since Kichuchi [29] sequenced and published the genome data of PWNs, many researchers have used transcriptome sequencing analysis to evaluate PWN molecular mechanisms, enhancing the knowledge of PWN. For example, PWN parasitic pine-related genes have been identified [30], revealing the interactions between PWNs and host plants. By comparing the transcriptomes, we can compare the genetic differences among PWNs with different virulence levels and *B. mucronatus* to identify virulence-related factors of PWNs, such as venom allergen-like proteins (VAPs) [31]. Transcriptome research has also significantly contributed to the genetic field of stress resistance and the growth and development of PWNs. By evaluating the PWN transcriptome, Feng and Li analyzed the response of PWNs to emamectin benzoate and the pine anti-infection substance α-pinene [32,33] to explain the process of detoxification during infection.

Introducing exogenous double-stranded RNA (dsRNA) can cause specific degradation of mRNA and lead to its corresponding gene silencing. This phenomenon is called RNAi [34]. In recent years, RNAi technology has been frequently used to assess PWN gene function, such as identifying the role of cathepsin L-like cysteine protease in PWN reproduction and pathogenicity [35], the role of *Bx-daf-22* in PWN mate attraction [36] and the role of *daf-16-2b* in PWN detoxification metabolism processes [37]. RNAi technology has also played an important role in the study of the low-temperature adaptation mechanism of PWNs. The low-temperature response mechanism of *Caenorhabditis elegans* was studied by RNAi. Researchers have found that this mechanism is regulated by many genes, including those associated with the cyclic guanosine phosphate (cGMP) pathway, cell autophagy pathway [38], and G protein signaling pathway [39], and have assessed the low-temperature response mechanism of PWNs. When the expression of the *daf-11* and *BxATG8* genes was inhibited, the survival and reproductive ability of PWNs at low temperatures decreased [40,41].

GD1 is a strain of bacteria isolated from *B. xylophilus* by Jiajin Tan in 2008 [42]. Previous studies showed that *Bacillus cereus* GD1 could significantly improve the survival rate of PWNs under low-temperature stress [43]. Similarly, at room temperature, GD1 can also improve the survival rate of pinewood nematodes, and there is an impact on the incidence of pinewood nematode disease. To understand the molecular mechanisms underlying how this strain improves the low-temperature adaptability of PWNs, PWNs were treated at low temperatures and the transcriptomes were sequenced. RNAi of the significantly differentially expressed gene *daf-11* verified its function in the low-temperature response by measuring the survival and reproduction rates, illustrating the role of associated bacteria in the low-temperature adaptability of PWNs.

## 2. Material and Methods

### 2.1. PWN Strain, Bacterial Strain, and Treatment

The nematode species *B. xylophilus* isolated from Anhui, China, named AH23, was used. The host was *Pinus massoniana*. The nematodes were cultured with *Botrytis cinerea* and stored at 4 °C. When the nematodes were removed, they were recultured for a generation at 25 °C. The associated bacterium GD1 was isolated from the body of PWNs by dilution spread on the LB medium after grinding the bodies of the nematodes. The bacterial strain was *Bacillus cereus*. Both the nematode and bacterial strains are preserved in the Forest Pathology Laboratory of Nanjing Forestry University (China). Before the experiment, all nematodes were disinfected with 0.1% streptomycin sulfate for 30 min and then rinsed 3 times with ddH_2_O.

Approximately 50,000 AH23 PWNs were mixed with 5 × 10^6^ cfu/mL GD1 bacterial suspension in the treatment group (T), and the same number of PWNs was mixed with the same amount of ddH_2_O in the control group (CK). T and CK had five repeat treatments each. All the PWNs were exposed to a temperature of 10 °C for 48 h. The supernatants were removed, and the pellet was washed 3 times with ddH_2_O, frozen in liquid nitrogen and stored at −80 °C.

### 2.2. Transcriptome Sequencing

The nematodes treated at 10 °C for 48 h were removed and RNA was extracted. The total RNA of PWNs was extracted by Trizol. After qualification by an Agilent 2100 instrument and agarose gel electrophoresis, the mRNA of nematodes was enriched by oligo (dt) magnetic beads. The fragmented cDNA library was synthesized and constructed by a New England Biolabs 7530 kit (Ipswich, MA, USA). The cDNA product was purified by 1.8× AgencourtAMPure XP Beads (Beckman Coulter, California, USA) and mixed with Illumina NovaSeq 6000 in the form of equimolar and paired terminal sequencing (PE150). RNA extraction, cDNA library construction and Illumina NovaSeq 6000 sequencing were completed by GeneDenovo Biotechnology Co., Ltd. (Guangzhou, China).

### 2.3. Transcriptome Data Filtering and Differential Gene Screening

Using the method of Chen, the original sequencing data obtained from Illumina NovaSeq 6000 were filtered using Fastp v 0.18.0 [44] to obtain high-quality clean reads. Using Bowtie (version 2.2.8) [45], the clean reads obtained in the previous step were compared with the ribosomal RNA (rRNA) database, the rRNA data were removed, and the clean reads were compared with the PWN genome (Wormbase: GCA_000231135.1) using HISAT v 2.2.4 [46]. Using the method of Pertea et al., StringTie (version 1.3.1) [47] was used to compare the assembly of clean reads to the whole genome, and the FPKM value (fragment per kilobase of transcript per million mapped reads) was calculated to eliminate the effect of different gene lengths and sequencing data on the calculation of gene expression levels. DESeq 2 [48] was used to analyze the differences between the control and transcriptional groups, and the genes with *p* value < 0.05 and log2| fold change | > 0.1375 were considered differentially expressed genes (DEGs).

### 2.4. Gene Ontology and KEGG Pathway Enrichment Analysis

Gene Ontology (GO) is often used to describe the properties of genes and their related products in organisms. We mapped differentially expressed proteins to each term in the GO database (http://www.geneontology.org/, accessed on 12 January 2021) and calculated the number of proteins per term to obtain the list of proteins with a certain GO function and number of proteins. A hypergeometric test was then applied to identify GO entries significantly enriched in differentially expressed proteins compared with the entire background protein. The *p*-value was calculated as follows:$$P=1\;-\overset{m-1}{\underset{i=0}{\sum }}\frac{\left(\begin{array}{c} M\\ i \end{array}\right)\left(\begin{array}{c} N-M\\ n-i \end{array}\right)}{\left(\begin{array}{c} N\\ n \end{array}\right)}$$

*N* is the number of proteins with GO annotations in all background proteins, *n* is the number of differentially expressed proteins in *N*, *M* is the number of proteins in all background proteins annotated for a specific GO term, *m* is the number of differentially expressed proteins annotated for a specific GO term. After the calculated *p*-value is corrected by Bonferroni, the threshold of the *p*-value ≤ 0.05 is defined as the GO term significantly enriched in the differentially expressed protein.

Due to the interaction between genes, some biological functions may be affected. To further study the function of genes, DEGs were enriched and analyzed by the KEGG database [49]. The *p*-value calculation formula for this hypothesis test is similar to the GO enrichment analysis. *N* is the number of pathway-annotated proteins in all background proteins, *n* is the number of differentially expressed proteins in *N*, *M* is the number of proteins in all background proteins annotated as a specific pathway, and *m* is the number of differentially expressed proteins annotated for a particular pathway. The calculation formula is as follows:$$P=1\;-\;\overset{m-1}{\underset{i=0}{\sum }}\frac{\left(\begin{array}{c} M\\ i \end{array}\right)\left(\begin{array}{c} N-M\\ n-i \end{array}\right)}{\left(\begin{array}{c} N\\ n \end{array}\right)}$$

### 2.5. qPCR Study of Transcriptome

To verify the accuracy of the DEGs in the transcriptome data, we randomly selected DEGs and designed specific primers with the NCBI online primer design tool (https://www.ncbi.nlm.nih.gov/tools/primer-blast/, accessed on 18 March 2021)) (Appendix A). qPCR was performed according to the manufacturer’s instructions (Vazyme, China). The accuracy of upregulation and downregulation of the DEGs was verified by qPCR using Applied Biosystems 7500 and 7500 Fast Real-Time PCR Systems (ABI, Los Angeles, CA, USA). Each gene has 3 biological repeats. The reference endogenous control gene was actin. The relative expression level of the genes in each group was calculated by the 2^−△△CT^ method, and the differences were analyzed by SPSS 24. Pearson’s correlation coefficient was used to verify the correlation of RNA-seq and qRT-PCR.

### 2.6. Daf-11 cDNA Fragment Cloning

Studies have shown that *daf-11* is related to the low-temperature adaptability of nematodes [43]; thus, we selected the differentially expressed gene *daf-11* to verify the effect of associated bacteria on the low-temperature adaptability of PWNs. Primers were designed (*daf-11*_F, *daf-11*_R) using primer design software Primer 5.0 and the blast *daf-11* CDS in the whole PWN genome as the template (Appendix A). The selected fragments were amplified by PCR using a Phanta Max Super-Fidelity DNA Polymerase kit (Vazyme, China). Next, the fragments were inserted into the pCE2-TA/Blunt-Zero vector and then were transformed into E. coli (DH5α) competent cells under the following conditions: 0 °C for 30 min, 42 °C for 45 s, 0 °C for 2 min. The samples were then incubated at 37 °C for 16 h, and then single colonies were selected for PCR (the primers were M13F and M13R). The PCR product bands were examined by agarose gel electrophoresis. After confirming the correct size of the bands, sequencing was performed by Shenggong Bioengineering Co., Ltd. (Shanghai, China). The sequencing results were compared with the reference genomic CDS by BioEdit to confirm that the gene fragment was cloned successfully.

### 2.7. dsRNA Synthesis of Daf-11

Through an online design website (Invitrogen Block-iT RNAi Designer (https://origin-k8s-prodb.cloudprod.thermofisher.com/rnaiexpress/setOption.do?designOption=sirna&pid=-197758385940235531 (accessed on 21 April 2021)), dsRNA sequences were designed using the *daf-11* CDS as templates and dsRNA interfering with *gfp* expression as a control. Four single-stranded oligoDNA sequences with T7 promoter sequences (underlined) were designed (Appendix A). The sequence preparation service was provided by GenScript Biology Co., Ltd. (Nanjing, China). *Ds_daf-11_1*, *ds_daf-11_2*, *ds_daf-11_3*, and *ds_daf-11_4* were paired to synthesize double-stranded Oligo DNA according to the manufacturer’s instructions (TaKaRa, China). It was used to synthesize *daf-11* dsRNA. *Gfp* dsRNA was synthesized using the same method. The dsRNA was stored at −80 °C.

### 2.8. Detection of Daf-11 Interference Efficiency

A total of 5000 PWNs were soaked in *daf-11* dsRNA, *gfp* dsRNA and RNase-free water and soaked at 20 ℃ for 48 h. The dsRNA was washed with RNase-free water. Then, PWNs were quickly frozen with liquid nitrogen. PWN RNA was extracted by Trizol, and a kit (Vazyme, China) was used to reverse-transcribe the extracted RNA to obtain cDNA. *gfp* dsRNA treatment was used as a dsRNA control (*si_gfp*), the same number of AH23s soaked in RNase-free water as a blank control (*WT*) and *daf-11* dsRNA treatment as the treatment (*si_daf-11*). The interference efficiency was detected by qRT-PCR, and three biological repeats were performed.

### 2.9. Effects of Daf-11 RNAi on the Low-Temperature Survival Rate and Fecundity of AH23

Approximately 6000 AH23 were soaked in *daf-11* dsRNA for 48 h and then placed at 10 °C for treatment (*si_daf-11*). The same number of AH23s were treated with *gfp* dsRNA as a dsRNA control (*si_gfp*), and wild pinewood nematodes soaked in ddH_2_O were used as a blank control (*WT*). The experiments were repeated three times. The 1-day survival rate was measured, and the survival rate was calculated every 2 days at least 5 times per day.

Approximately 500 AH23s treated with *daf-11* dsRNA were inoculated with *Botrytis cinerea* as a treatment (*si_daf-11*), with the same number of AH23s treated with *gfp* dsRNA as the control (*si_gfp*) and the same number of AH23s soaked in ddH_2_O as the blank control (*WT*). The nematodes were placed at 15 °C, and the experiment was repeated 5 times (because the propagation behavior of pinewood nematodes at 10 °C is slow, which is not conducive to experimental data collection, 15 °C was used). After 15 days, the nematodes were collected and reproduction levels measured.

### 2.10. Data Statistics

All the data are expressed as means ± standard error of the mean (SEM). All the parameters were calculated using Microsoft Excel 2019. Student’s *t*-test was performed using IBM SPSS Statistics 24. Differences were considered statistically significant at *p* < 0.05.

## 3. Result

### 3.1. Transcriptome Data Filtering

A total of 542,039,484 reads were obtained by transcriptome sequencing. Fastp showed that 540,810,946 clean reads were obtained, and the proportion of clean reads was over 99.7%. The GC content of the sequence was 47.59–47.97%. HISTA2 showed that the match between clean reads and the reference genome was over 90.68%. Thus, the sample quality was sufficient for the follow-up analysis and study (Table 1). The comparison with the PWN ribosomal database revealed that 0.05–0.1% clean reads mapped to rRNA. The unmapped reads were reserved for subsequent analyses (accession number to cite the SRA data: PRJNA794276).

### 3.2. Differentially Expressed Gene (DEG) Analysis

As shown in Table 2, 16,398 genes were detected, of which 16,082 genes matched the reference genome, accounting for 90.84%. In sum, 316 new genes were detected, indicating that the selected reference genome was relatively perfect.

Using a *p*-value < 0.05 and log2|fold change| > 0.138 as the standard, 439 DEGs were obtained between T and CK, of which 206 were upregulated and 233 downregulated (Figure 1A). The gene expression levels between biological repeats were similar, but there was a significant difference between T and CK (Figure 1B). These genes may play a key role in the low-temperature adaptability of AH23.

### 3.3. Transcriptome Data Verification by qRT-PCR

To verify the transcriptome data, 10 DEGs were randomly selected, and the qRT-PCR data were consistent with the RNA-seq data (Figure 2). Pearson’s correlation coefficient was 0.916, and the qRT-PCR results correlated significantly with RNA-seq. This showed that the RNA-seq data were reliable and could be used to screen genes related to low-temperature adaptability.

### 3.4. Gene Ontology (GO) and KEGG Pathway Enrichment Analysis

The GO results showed that the DEGs were classified into 48 GO terms (Figure 3). Biological process (BP) was the most annotated, accounting for 48% of the total, among which 106 DEGs were involved in single biological process, 109 DEGs were involved in cell process, 83 DEGs were involved in metabolic process, 65 DEGs were involved in biological regulation, 48 DEGs were involved in response to stimulus, and 45 DEGs were involved in developmental process. Molecular function (MF) annotation accounted for 19% of the total, of which 72 DEGs were related to binding and 67 DEGs were related to catalytic activity. Cell components (CC) annotation accounted for 33% of the total, among which 85 DEGs were related to cell part and 63 DEGs were related to organelle. In addition, some genes were annotated that related to biological growth processes and reproductive processes, which may be related to low-temperature adaptability. Through GO enrichment analysis, 1585 GO terms were enriched, of which 102 were significantly enriched (*p* < 0.05) (Figure 4). These significantly enriched GO terms may help investigate the low-temperature adaptation process of PWN.

KEGG was used to dissect the key genes to improve the low-temperature adaptability of pinewood nematodes. All the DEGs were analyzed in the KEGG database, and 207 pathways were annotated (Figure 5, Appendix A). Among them, apoptosis, mucin type O-glycan biosynthesis, sphingolipid signaling pathway, and alanine, aspartate and glutamate metabolism were significantly enriched (*p* < 0.05) (Figure 6). GD1 may have an effect on the energy metabolism and intracellular components of AH23. Additionally, GD1 also affected sphingolipid metabolism, glutathione metabolism, arachidonic acid metabolism, ether lipid metabolism, glycerophospholipid metabolism, inositol phosphate metabolism, and cysteine and methionine metabolism of AH23. Thus, GD1 may affect the fat composition, cell membrane structure and energy metabolism of AH23. These pathways are mainly concentrated in three categories: metabolism, cellular processes and environmental information processing (Figure 6). This shows that the improvement of the low-temperature adaptability of AH23 as a result of GD1 may occur through energy metabolism, cell membrane structure and inclusion component changes. This provided a reference for our follow-up screening of low-temperature-related genes. KEGG pathway analysis revealed significant changes in genes related to nematode reproduction, longevity and environmental stress, such as *lin-45*, *ife-3*, *gld-8*, *SconB*, *MTM-6*, *SNF-8*, *FAAH2* and *PHPT1*. Additionally, many genes associated with the differentiation and metabolism of nematode fat also underwent significant changes, such as *Taz*, *sae-1*, and *erp44*. We screened the key gene *daf-11* for the low-temperature adaptation of pinewood nematodes in the ribose metabolism pathway.

### 3.5. Fragment Clone of Daf-11

To date, many studies [41] have revealed genes related to the low-temperature adaptability of PWNs. In this study, it was found that the *daf-11* gene, which is related to the low-temperature adaptability of PWNs, was significantly upregulated by GD1. The relative expression of the treatment group (T) was 2.5 times that of the control group (CK) (Figure 7A). Thus, we designed specific primers for PCR amplification of *daf-11* and sequenced it after agarose gel electrophoresis. The results showed that *daf-11* was expressed in AH23 (Appendix A), and the length of the product was 1368 bp. After alignment with the genome sequence of PWN, the sequencing result was consistent with that of the *daf-11* CDS.

### 3.6. RNAi of Daf-11

The results showed that the expression of *daf-11* in those treated with *si_daf-11* was significantly lower than that in the control (Figure 7B): the expression was only 33.3% of that in the latter. Therefore, *daf-11* dsRNA can be used for RNAi to verify the effect of *daf-11* on the survival and reproduction of PWNs at low temperatures.

Survival and reproduction have been selected by many researchers as indicators of the temperature adaptability of nematodes [10]. In this study, changes in the survival and reproduction capabilities of AH23 in a low-temperature environment were observed by RNAi of *daf-11*. The results showed that when *daf-11* was silenced, the survival rate and reproduction abilities of AH23 at low temperatures were impaired compared with those of the control. At 10 °C, the survival rate of *si_daf-11* was significantly lower than that of the control from the 4th day, and the survival gap widened over time, indicating that that *daf-11* could improve the survival rate of AH23 at low temperatures (Figure 7C). The results also showed that when *daf-11* was silenced, the reproduction ability of AH23 was significantly reduced at 15 °C compared with that of the control. After 15 days of culture, the number of *si_daf-11* nematodes was 2190, which was significantly lower than the numbers in the controls (5640 and 5800) (Figure 7D). This shows that *daf-11* can significantly improve the reproductive ability of AH23 in a low-temperature environment.

## 4. Discussion

PWD is a destructive disease that affects pine worldwide and has caused inestimable ecological losses to many countries in Asia [50]. Because of its continuous spread to lower-temperature areas, the important influencing factors during PWD spread and its epidemiology must be analyzed. To date, however, little is known about the genetic mechanism of how PWNs respond to low temperatures. In recent years, an increasing number of studies have shown that the bacteria associated with PWNs can improve their ecological adaptability [50].

The low-temperature response of animals is a regulated genetic process [40]. Some studies have shown that the environmental response-related genes of animals can be screened by RNA-seq in different ecological environments. We previously found that the low-temperature adaptability of PWNs was significantly improved under the action of GD1, which was reflected in the improvement in the low-temperature survival rate and reproduction ability, but the mechanism by which GD1 improved the low-temperature adaptability of PWN remained unclear. In this paper, the effect of GD1 on the expression of low-temperature response genes in PWNs was studied by RNA-seq. The results showed that GD1 regulated the low-temperature adaptation of PWNs through various genes. Some studies have shown that PWNs transform into diffuse third instar larvae during low-temperature adaptation, greatly improving their survival ability. This transformation is often accompanied by the thickening of the body wall and the accumulation of inclusions [10]. Because of their low melting point and role in maintaining the fluidity of the cellular lipid membrane in a low-temperature environment, fat and fatty acids are often related to low-temperature adaptability. In the process of low-temperature adaptation by PWNs, the content of fatty acids in the body significantly increases, especially unsaturated fatty acids, which is often positively correlated with the low-temperature survival rate of PWNs [10]. Through KEGG analysis, it was found that there were differences in sphingolipid metabolism, arachidonic acid metabolism, glycerophospholipid metabolism and inositol phosphate metabolism after treatment with GD1. This finding is in accordance with the results of previous studies of these pathways [51,52,53,54]. It is suggested that GD1 affects the low-temperature adaptability of PWNs by energy metabolism and cell component levels, especially fatty acid components. qRT-PCR is widely used to verify the accuracy of RNA-seq [55]. The different calculation methods of RNA-seq and qRT-PCR may lead to different degrees of differences in expression results, as frequently observed in the literature [55]. Here, the qRT-PCR verification results were associated with the RNA-seq results using Pearson’s correlation coefficient, which proved the reliability of the RNA-seq results.

Thermotaxis is highly significant to the transmission and spread of PWNs [9]. Since researchers identified the thermotaxis capability of *C. elegans* in 1975 [56], an increasing number of studies have been performed on the temperature perception of nematodes. Chemosensors and related neurons play a critical role in the low-temperature response of *C. elegans* [57], and many genes are involved in this process. *Daf-11* can encode receptor guanylate cyclase, which plays a key role in the molecular regulation of the G protein signal pathway and cGMP pathway involved in low-temperature adaptation. *Daf-11* can regulate the physiological activity of nematodes by adjusting the ion concentration in the cytoplasm [39]. Wang identified and verified the low temperature-related gene *daf-11* [43] by homologous comparison of the genome of *C. elegans*. In this paper, the expression of *daf-11* in AH23 was verified by designing specific primers. The results of RNA-seq and qRT-PCR showed that GD1 treatment significantly increased the expression of *daf-11* in PWNs at low temperatures. RNAi further verified that the *daf-11* gene was related to the survival rate and reproduction ability of PWNs at low temperatures. However, the upregulated expression of *daf-11* was due to GD1. Therefore, the author speculates that GD1 could positively regulate the expression of the *daf-11* gene in PWNs to change the ion concentration in the cytoplasm, thus regulating the physiological activities of PWNs to improve their ability to adapt to low temperature. Thus, a mechanism exists between GD1 and PWN: GD1 improves the low-temperature adaptation ability of nematodes by regulating the expression of low-temperature adaptation genes of PWN.

The results of this research showed that the associated bacteria GD1 affected the low-temperature adaptability of PWNs. As important microorganisms that interact with PWNs in nature, bacteria are critical for their ecological adaptability. The bacterial consortia found in the cuticular layer of PWNs in Europe may help the nematodes withstand toxic terpenes, benzoic acid, resin acid and other substances in the host. *Serratia* spp. isolated from the body surface of PWNs can improve their antioxidant capacity [50]. Bacteria may affect the feeding and spreading of PWNs through horizontal gene transfer (HGT) [50]. In addition to *daf-11*, other differentially expressed genes related to environmental stress and fat metabolism have been identified by RNA-seq analysis, such as *SconB*, *FAAH2* and *PHPT1* [58,59,60]. Energy and material metabolism are important means for animals to adapt to low temperatures. This study shows that substance metabolism, especially fatty acid metabolism, by PWNs changed significantly under GD1 treatment, which provides ideas for follow-up studies. At present, the functions of these differential genes in the low-temperature adaptation of PWN have not been reported and are worthy of further study. This study preliminarily discussed the effect of the associated bacteria GD1 on the low-temperature adaptability of PWNs and provides new ideas for elucidating the mechanism by which PWNs alter the temperatures at which they are viable and spread to low-temperature areas.

## Figures and Tables

**Figure 1 microorganisms-11-00430-f001:**
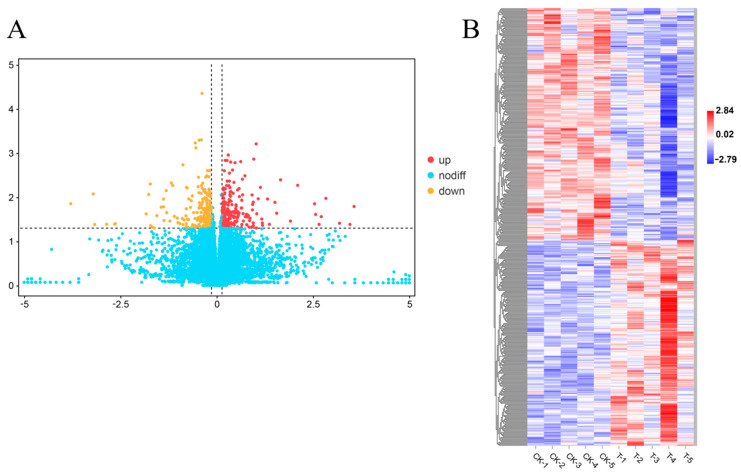
DEG and expression of transcriptome. (**A**) The number of DEGs. Yellow dots represent the significantly downregulated genes. Red dots represent the significantly upregulated genes. (**B**) Expression of DEGs. Red represents upregulated expression. Blue represents downregulated expression.

**Figure 2 microorganisms-11-00430-f002:**
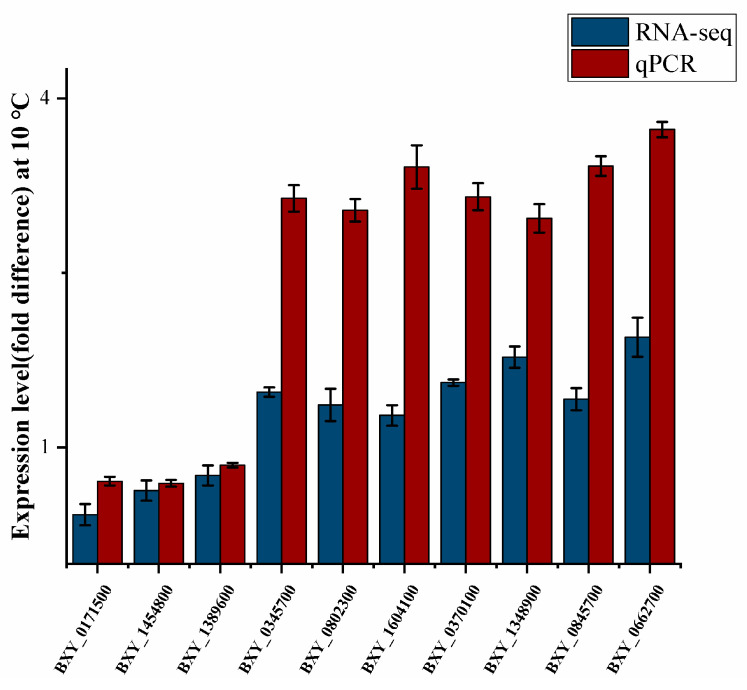
Quantitative real-time polymerase chain reaction (qRT-PCR) identification of differentially expressed DEGs between CK and T group nematodes. The relative gene expression levels were normalized to the internal control gene actin.

**Figure 3 microorganisms-11-00430-f003:**
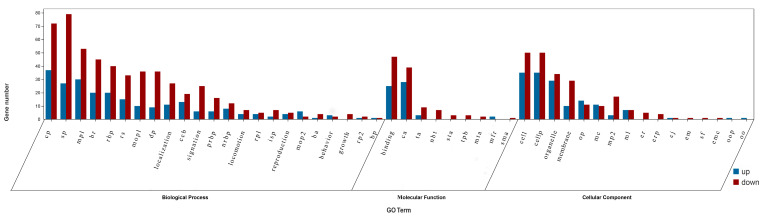
Gene Ontology (GO) categories of DEGs between CK and T group nematodes. The terms represented by the acronyms are as follows: cp, cellular process; sp, single-organism process; mp1, metabolic process; br, biological regulation; rbp, regulation of biological process; rs, response to stimulus; mop1, multicellular organismal process; dp, developmental process; ccb, cellular component organization or biogenesis; prbp, positive regulation of biological process; nrbp, negative regulation of biological process; rp1, reproductive process; isp, immune system process; mop2, multiorganism process; ba, biological adhesion; rp2, rhythmic process; bp, biological phase; ca, catalytic activity; ta, transporter activity; nbt, nucleic acid binding transcription factor activity; sta, signal transducer activity; tpb, transcription factor activity protein; mta, molecular transducer activity; mfr, molecular function regular; sma, structural molecule activity; cellp, cell part; op, organelle part; mc, macromolecular complex; mp2, membrane part; ml, molecular transducer activity; mfr, molecular function regular; smal region activity; cellp, cell part; op, organelle, matrix part; mc, macromolecular component; mp2, extracellular component.

**Figure 4 microorganisms-11-00430-f004:**
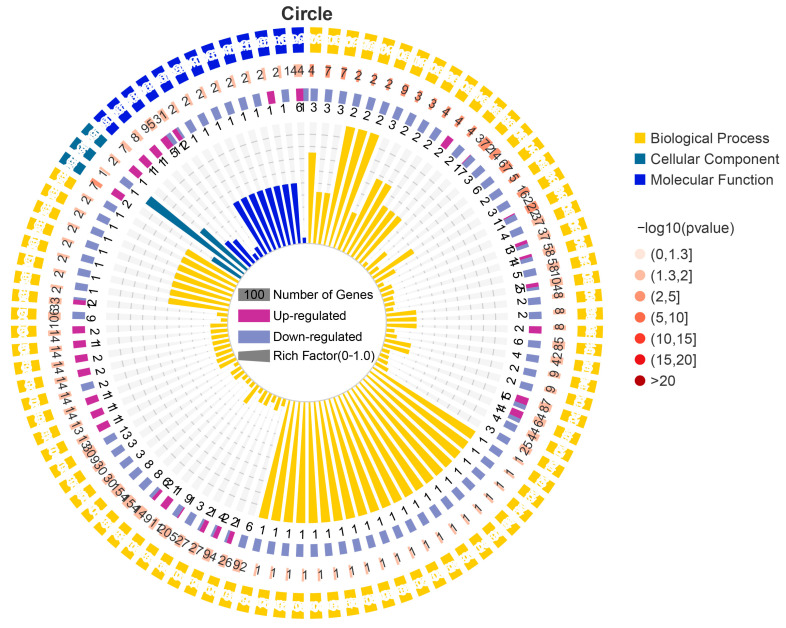
Circle of GO enrichment. First Circle (Outermost Circle): first 100 enriched GO terms, with different colors representing different ontologies. Second circle: number of genes in the GO term and *p* value. The higher the number of genes, the longer the strips, the smaller the *p*-value, and the redder the color. The third circle: up-and-down gene ratio bar chart; dark purple represents the up-regulation gene ratio, and light purple represents the downregulation gene proportion; the specific values are displayed below. Fourth round (innermost circle): rich factor value (number of DEGs in the GO term divided by the number of background genes in the GO term).

**Figure 5 microorganisms-11-00430-f005:**
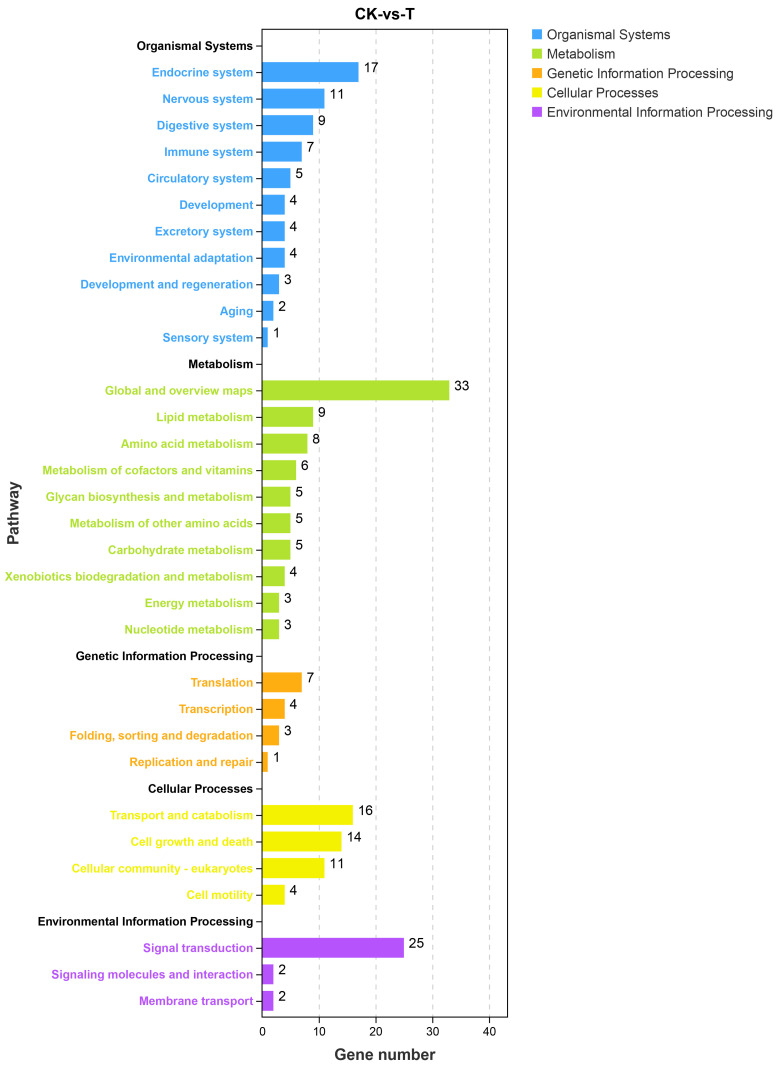
KEGG analysis of DEGs. The ordinate represents the pathway, and the abscissa represents the number of genes.

**Figure 6 microorganisms-11-00430-f006:**
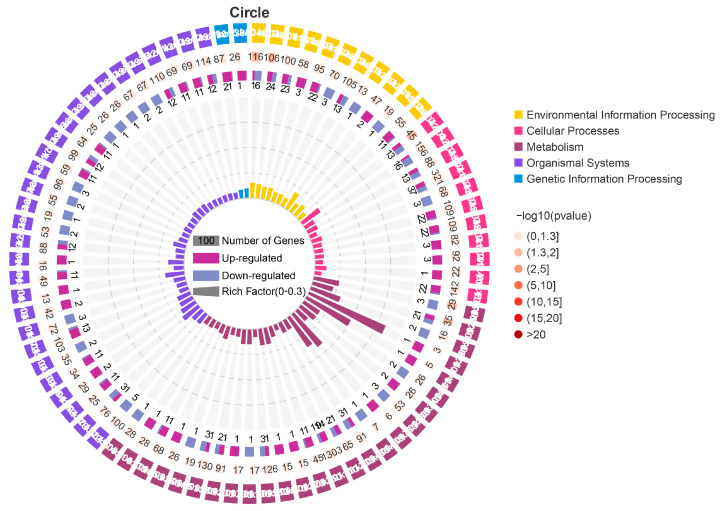
Circle of KEGG enrichment. First circle (outermost): first 100 enriched pathways, with different colors representing different KEGG A Class. Second circle: number of genes in the pathway and *p* value. The higher the number of genes, the longer the strips, the smaller the *p* value, and the redder the color. The third circle: up-and-down gene ratio bar chart; dark purple represents the upregulation gene ratio, and light purple represents the downregulation gene proportion; the specific values are displayed below. Fourth round (innermost circle): rich factor value (the number of DEGs in the pathway divided by the number of background genes in the pathway).

**Figure 7 microorganisms-11-00430-f007:**
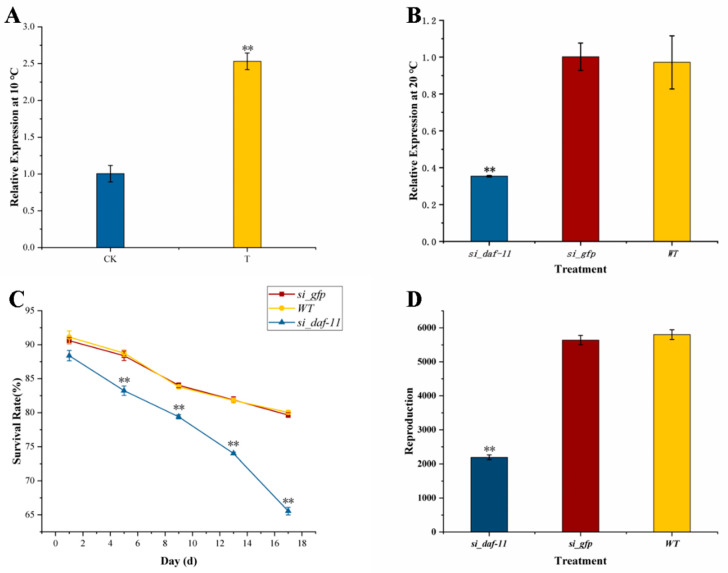
RNAi of *daf-11*. (**A**) Relative expression of *daf-11*. CK is the PWN sample treated with sterile water; T is the PWN sample treated with GD1. The ordinate coordinate represents the relative *daf-11* expression. (**B**) The RNAi efficiency of *daf-11* after socked at 20 °C. *si_daf-11* is the *daf-11* dsRNA treatment, *si_gfp* is the *gfp* dsRNA control, and *WT* is the wild-type blank control. (**C**) Survival rate of AH23 at 10 °C after RNAi. (**D**) Reproductive capacity of AH23 at 15 °C after RNAi. Date represent mean values ± SD from different repetitions. Asterisks indicates statistically significant differences (** *p* < 0.01).

**Table 1 microorganisms-11-00430-t001:** Transcriptome data filtering statistics.

Transcriptome Date	
Raw reads (pair)	542,039,484
Clean reads (pair)	540,810,946
Raw nucleotide length (bp)	81,305,922,600
Mapped reads (%)	0.05–0.1%
Unique mapped (%)	90.68–91.30%

**Table 2 microorganisms-11-00430-t002:** Statistics of transcriptome gene detection.

Sample	Refer Genes (%)	Novel Genes (%)	Total Genes (%)
All	16,082 (90.84%)	316 (100.00%)	16,398 (91.00%)
CK1	14,972 (84.57%)	289 (91.46%)	15,261 (84.69%)
CK2	14,996 (84.70%)	289 (91.46%)	15,285 (84.82%)
CK3	15,013 (84.80%)	290 (91.77%)	15,303 (84.92%)
CK4	14,969 (84.55%)	289 (91.46%)	15,258 (84.67%)
CK5	15,272 (86.26%)	294 (93.04%)	15,566 (86.38%)
T1	15,045 (84.98%)	292 (92.41%)	15,337 (85.11%)
T2	15,088 (85.22%)	294 (93.04%)	15,382 (85.36%)
T3	14,972 (84.57%)	291 (92.09%)	15,263 (84.70%)
T4	15,228 (86.01%)	300 (94.94%)	15,528 (86.17%)
T5	15,009 (84.78%)	284 (89.87%)	15,293 (84.87%)

## Data Availability

The datasets generated and analyzed during the current study are available in the Sequence Read Archive (SRA) repository, https://dataview.ncbi.nlm.nih.gov/object/PRJNA794276?reviewer=4nqnchkm2jcan7mapbf94c5lnv (accessed on 5 January 2022).

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
