# Peer review of "Effect of Associated Bacteria GD1 on the Low-Temperature Adaptability of Bursaphelenchus xylophilus Based on RNA-Seq and RNAi"

_microorganisms, 2023, doi:10.3390/microorganisms11020430_

Round 1

Reviewer 1 Report

The paper presents some very intersesting results and is a continuation of previous work relating to the pssible role  of bacteria associated with the nematode Bursaphelenchus xylophilus. The results are novel and explains in great detail have the bacterial strain GD1 plays a role in the climate adaptation. The results are very importatant as new outbreaks of B. xylophilus have been found in colder areas which origional not thought to have any problems. The adaptation of the PWN to colder climates is extremeley important and has world wide consequences. 

The transcriptome data is well presented and the verifictation using qRT-PCR is correct. The daf-11 gene which has previously been shown to be inportant in adaptation in other experimental systems was over expressed when the nematodes were treated with GD1, therefore givs more evidence of the role of GD1 in cold adaptation. 

This paper has opened a new line of investigation which should be encouraged. What are the other roles of the other asociated bacteria.? I think that the article has a wide audience and should be published.

Author Response

Thank you very much for your comments and suggestions. We also found that the strain GD1 can improve host adaptability of PWN, but results have not been published.

Reviewer 2 Report

I have read several times this paper. The authors follow a cannonical and academic approach and the experiments are in agreement with the thesis they have focussed. In fact they have demonstarted that daf-11 can improve reproductivity ability of Bursaphelenchus xylophilus at low temperature. I did not find any reason for not publishing it in Microorganisms. My recommendation is to accept the paper in its present form.

But, please:

page 2, line 92......add date, reference or deposit collection about GD1 isolation by Jianjin Tan.

page 12, line345......change de paragraph number label (It must be 3.6 instead 3.7)

Author Response

Thank you very much for your comments. I agree with your suggestions and have revised them in red. I hope that these revisions will be recognized.

Point1: page 2, line 92......add date, reference or deposit collection about GD1 isolation by Jianjin Tan.

Response1: I have added the date and reference about GD1 isolation by Jiajin Tan in the manuscript.

Point2: page 12, line345......change de paragraph number label (It must be 3.6 instead 3.7)

Response2: I apologize to you for my mistake and I have revised it in my manuscript.
